# Changes in Lean Tissue Mass, Fat Mass, Biological Parameters and Resting Energy Expenditure over 24 Months Following Sleeve Gastrectomy

**DOI:** 10.3390/nu15051201

**Published:** 2023-02-27

**Authors:** Laurent Maïmoun, Safa Aouinti, Marion Puech, Patrick Lefebvre, Melanie Deloze, Pascal de Santa Barbara, Eric Renard, Jean-Paul Christol, Justine Myzia, Marie-Christine Picot, Denis Mariano-Goulart, David Nocca

**Affiliations:** 1Département of Nuclear Medicine, Hôpital Lapeyronie, CHU de Montpellier, 34295 Montpellier, France; 2PhyMedExp, Université de Montpellier, INSERM, CNRS, 34090 Montpellier, France; 3Unité de Recherche Clinique, Biostatistiques et Epidémiologie, Département de l’Information Médicale, CHU de Montpellier, 34295 Montpellier, France; 4Service de Chirurgie Digestive A, Hôpital Saint Eloi, CHU de Montpellier, 34295 Montpellier, France; 5Departement d’Endocrinologie, Diabetes, Nutrition, Hôpital Lapeyronie, CHU de Montpellier, 34295 Montpellier, France; 6Département de Biochimie, Hôpital Lapeyronie, CHU de Montpellier, 34295 Montpellier, France; 7Département de Physiologie Clinique, Université de Montpellier, Hôpital Lapeyronie, CHRU de Montpellier, 34295 Montpellier, France

**Keywords:** sleeve gastrectomy, fat mass, lean tissue mass, visceral adipose tissue, IGF-1, IGFBP-3, resting energy expenditure, c-reactive protein, Hba1c, HOMA

## Abstract

Sleeve gastrectomy (SG) induces weight loss but its effects on body composition (BC) are less well known. The aims of this longitudinal study were to analyse the BC changes from the acute phase up to weight stabilization following SG. Variations in the biological parameters related to glucose, lipids, inflammation, and resting energy expenditure (REE) were concomitantly analysed. Fat mass (FM), lean tissue mass (LTM), and visceral adipose tissue (VAT) were determined by dual-energy X-ray absorptiometry in 83 obese patients (75.9% women) before SG and 1, 12 and 24 months later. After 1 month, LTM and FM losses were comparable, whereas at 12 months the loss of FM exceeded that of LTM. Over this period, VAT also decreased significantly, biological parameters became normalized, and REE was reduced. For most of the BC, biological and metabolic parameters, no substantial variation was demonstrated beyond 12 months. In summary, SG induced a modification in BC changes during the first 12 months following SG. Although the significant LTM loss was not associated with an increase in sarcopenia prevalence, the preservation of LTM might have limited the reduction in REE, which is a longer-term weight-regain criterion.

## 1. Introduction

Bariatric surgery (BS) is an effective method for both acute and long-term weight loss in obese patients when other treatments have failed and when the body mass index (BMI) is greater than 40 kg/m^2^ (severe obesity) or greater than 35 kg/m^2^ with obesity-related comorbidities [1]. The loss of excess body weight is generally a criterion for successful surgery. However, while the change in absolute weight loss provides an indication of progress [2] and is easy to use in clinical practice, it cannot adequately reflect changes in fat mass (FM) and lean tissue mass (LTM), the two compartments that constitute body composition. These two indicators are more closely related to all-causes and cardiovascular mortality than to weight [3].

After BS, the greatest weight loss occurs in the first year and weight stabilization is achieved in the second [4,5]. Weight regain may start after the second or third year [4,5]. Conversely, the long-time course of body composition change is unclear, probably because the techniques of investigation—including bioelectrical impedance analysis (BIA) and dual-energy X-ray absorptiomety (DXA)—may introduce measurement bias [6]. Moreover, although BIA is less expensive than DXA, it may underestimate FM and overestimate LTM in patients with obesity [7]. The body hydration disturbance present in this population may lead to measurement errors [8]. When the reference technique is used (i.e., DXA) [9], a biphasic variation is observed. This has been characterized by an acute and concomitant reduction in FM and LTM in the first few months after BS, followed by a sustained reduction in FM during the subsequent prolonged weight-loss period [10,11,12]. Generally, the time of follow-up in these studies has been limited to the first 12 months [11,12,13], and very few longitudinal studies have been performed between 24 and 36 months [14,15]. Yet monitoring body composition change over a longer postsurgery period may provide clinically relevant information to better identify and treat patients according to lifestyle and medical care [13,16]. Moreover, although it is not clear whether body composition changes vary with the surgical procedure, studies have preferentially focused on the Roux-en-Y gastric bypass (RYGB) technique rather than sleeve gastrectomy (SG) [13], despite SG recently becoming the most common bariatric approach [17].

It was also reported that patients with weight regain presented higher %FM and lower %LTM than patients maintaining stable weight [18]. Moreover, a lower weight-adjusted resting energy expenditure (REE) was observed in the weight regain group [18], suggesting that this clinical parameter should also be routinely evaluated. These studies, thus, seem to indicate that an excessive reduction in LTM during weight loss programs may have some deleterious effects on metabolism, thermoregulation, functional capacity and weight regain [19].

While body composition analysis appears to be an improvement over simple body weight assessment, the measurement of visceral adipose tissue (VAT) may be even more relevant in patients with obesity. For example, VAT excess induces and maintains lipotoxicity and insulin resistance [20,21], playing a central role in cardiac dysfunction [22]. VAT is also an important endocrine organ that secretes pro-inflammatory factors, resulting in chronic low-grade systemic inflammation that may be involved in the pathogenesis of metabolic abnormalities [23]. However, to the best of our knowledge, few studies have investigated the VAT change after BS [24,25].

The aims of this study were to analyse the FM, LTM and VAT changes from the acute phase of body weight loss (1 month) until a recognized phase of body weight stabilization (12 and 24 months) following SG. We also analysed the variation in the biological parameters related to glucose, lipids and inflammation, as well as the resting energy expenditure.

## 2. Materials and Methods

### 2.1. Subjects and Method

The study was approved by the local research ethics committee (ID RCB: 2015-A01047-42). All patients signed a consent form before entering the study. The clinical trial number is NCT02712086.

#### 2.1.1. Subjects

From November 2016 to July 2020, 83 Caucasian patients (women *n* = 63; 75.9%) from 18.4 to 60.0 years old were recruited from candidates for BS in the obesity management centre, University Hospital of Montpellier (Montpellier, France). The inclusion criteria were: inaction of other weight loss treatments, BMI > 40 or BMI ≥ 35 kg/m^2^ with the presence of comorbidities [type 2 diabetes (T2D), sleep apnoea syndrome or arterial hypertension (HTA)] and more than 4 years of obesity [1]. The exclusion criteria were: previous BS, pregnancy, medical treatment or physical handicap that might affect body composition evaluation, and body weight >190 kg or height >192.5 cm (limitations of the densitometry device). Physical activity levels were not specifically determined, but none of the participants was participating in a training program in the period before inclusion. Medical history was obtained by questionnaires. All the BS procedures were sleeve gastrectomy (SG), which consists of resecting most of the greater curvature to reduce gastric size and leaving a narrow stomach tube. The SGs were performed in a single institution and in only one surgical department.

#### 2.1.2. Methods

This study followed a longitudinal design; its methodology was previously described in detail [10,11]. However, the data of the patients included in this study have never been published. Briefly, all the patients were evaluated the day before the operation (baseline) and 1, 12 and 24 months after SG. After SG, patients were encouraged to increase their physical activity, improve protein intake, and reduce fat intake in order to lose weight, while avoiding side effects, such as muscle mass loss or steatorrhea. For each visit, standing height and weight were measured with a stadiometer and a weight scale with a precision of 0.1 kg. BMI was determined as weight (kg)/height^2^ (m). The ideal body weight (IBW in kg) was obtained from the Lorentz equations to calculate ideal body weight (IBW): (height [cm]—100)—((height [cm]—150)/4) for men and (height [cm]—100)—((height [cm]—150)/2.5) for women.

#### 2.1.3. Comorbidity Definitions

T2D was defined as glycated haemoglobin (HbA1c) ≥6.5% and/or fasting glycaemia ≥7 mmol/L and/or antidiabetic treatment [26].Arterial hypertension (HTA) was defined as systolic blood pressure >140 mmHg and/or diastolic blood pressure >90 mmHg and/or use of antihypertensive medications [27].

#### 2.1.4. Regional and Whole Body FM and LTM

FM (kg, %) and LTM (kg) were measured using DXA (Horizon A, Hologic, Inc., Waltham, MA, USA) at whole body and regional sites (upper limbs, trunk, and lower limbs). The same operator performed all scanning and analyses to ensure consistency after following standard quality control procedures. For LTM and FM, the coefficients of variation (CVs) given by the manufacturer were <1%. In the abdominal region, total adipose tissue (TAT), VAT and superficial adipose tissue (SAT) were measured according to a previously validated method [28].

To define sarcopenia in terms of low LTM, we chose the most current cut-offs used for the Caucasian population. The sum of the LTM of the arms and legs defined the appendicular lean mass (ALM; kg). ALM/height^2^ index [ALMI(h^2^); kg/m^2^] or ALMI/body mass index [ALMI(BMI)] defined the ALM index. ALM < 20 kg and ALMI(h^2^) < 7 kg/m^2^ in men and ALM < 15 kg and ALMI(h^2^) < 5.5 kg/m^2^ in women [29] defined sarcopenia according to The European Working Group on Sarcopenia in Older People (EWGSOP2). ALM < 19.75 kg and ALMI(BMI) < 0.789 in men and ALM < 15.02 kg and ALMI(BMI) <0.512 in women defined the cut-points for low LTM for sarcopenia according to the Foundation for the National Institutes of Health (FNIH) [30]. Finally, ALMI(h^2^) ≤ 7.23 kg/m^2^ in men and ≤5.67 kg/m^2^ in women [31] defined sarcopenia according to the International Working Group on Sarcopenia (IWGS, Albuquerque, NM, USA).

#### 2.1.5. Assays

Blood samples (35 mL) were collected in fasting conditions in the morning (8:30–9:00 am). The samples were centrifuged at 2500× *g* for 10 min at 4 °C. Serum samples were stored at −80 °C and most analyses were performed at the end of the study to reduce interassay variation. In premenopausal women, the blood samples were obtained at an unsynchronized menstrual stage.

Albumin, HbA1c, cholesterol, HDL, triglyceride, glucose, insulin and CRP were routinely analysed by Cobas 101, 501, 602 or 701 (Roche Diagnostic, Mannheim, Germany). The interassay and intraassay coefficients of CVs for the majority of these parameters were lower than 5%. The IGF-1 (Reference IS-3900) and IGFBP-3 (Reference IS-4400) assays are based on chemiluminescence technology and were analysed with IDS-iSYS (Immunodiagnostic Systems, Tyne & Ware, Boldon, UK). The intraassay and interassay CVs were 1.9 and 3.9% for IGF-1 and 1.8 and 6.3% for IGFBP-3. For all the biological parameters, the CVs for the intraassay and interassay variations were given by the manufacturer.

Insulin resistance was estimated using the homeostasis model assessment of insulin resistance (HOMA-IR) according to the following formula: fasting serum insulin (mIU/mL)/fasting plasma glucose (mmol/L)/22.5 [26].

#### 2.1.6. Resting Energy Expenditure (REE)

Measured REE (REE_m_) was performed in all patients between 8:00 and 8:30 h after overnight fasting and without the practice of physical exercise for 48 h. REEm was assessed by indirect calorimetry (Quark RMR, Cosmed, Rome, Italy), which analysed oxygen, carbon dioxide and ventilation in a mixing chamber over a period of at least 30 min. Predicted REE values (%; REE_p_) were calculated from the equation of Harris and Benedict modified by Roza and Shizgal [32] as follows: REE_p_ = 667.051 + 9.74 × (weight) + 1.729 × (height) − 4.737 × (age).

#### 2.1.7. Statistical Analysis

Quantitative variables are described with means and standard deviations (SD) after normality testing of the continuous variables with Shapiro–Wilk’s test. For categorical variables, the numbers and associated percentages are presented.

Paired Wilcoxon or paired Student’s tests were used, depending on the normality of the distribution, to compare the relative variations (100 × (measure 2 − measure 1)/measure 1) between baseline and 1, 12 and 24 months for the patients’ clinical characteristics; whole-body composition; android, gynoid, and abdominal adipose tissue; and biological parameters.

Relationships between preoperative characteristics (age, BMI, whole-body LTM, FM, TAT, VAT, SAT and biological parameters) and baseline or relative variations in body composition parameters at 1, 12 or 24 months were assessed using Spearman or Pearson correlation coefficients, depending on the normality of the distributions.

All analyses were two-tailed, with a *p*-value of < 0.05 considered statistically significant. SAS^®^ Enterprise Guide software (version 8.2, SAS Institute, Cary, NC, USA) was used to perform the analyses and graphs were generated using R statistical software (www.r-project.org, version 4.1.3) with ggplot2 package (version 3.3.3). 

## 3. Results

### 3.1. Anthropometric Parameters

A total of 83 patients with obesity underwent preoperative and at least one postoperative assessment of body composition (*n* = 83 with 1-month data; *n* = 76 with 12-month data. and *n* = 60 with 24-month data). The major reasons why 23 of the 83 subjects did not complete the 24-month follow-up were the following: they chose not to repeat the measurements (*n* = 15), were pregnant (*n* = 3), were relocated for work (*n* = 3), could not be located for follow-up (*n* = 2), or died (*n* = 1) (Table 1).

All the baseline and variations in anthropometric characteristics and comorbidity prevalence are presented in Table 1 and Figure 1. At inclusion, the mean age was 40.9 ± 12.3 years, and the mean BMI was 40.7 ± 4.2 kg/m^2^. The relative mean weight loss was −9.1 ± 2.1% after 1 month, −29.3 ± 8.4% after 12 months, and −27.5 ± 9.6% after 24 months. As expected, all the anthropometric parameters were lower at all times compared to baseline, but no additive loss was observed between 12 and 24 months. All the comorbidities decreased with time. Specifically, T2D was only present in six patients (<10%) at 24 months. Among the 17 patients who presented T2D at baseline, only two were lost to follow-up (one had died).

### 3.2. Body Composition

Initial values and changes in FM and LTM are presented in Table 2 and Figure 1. At 1 month, the reduction in % relative variation at the different sites ranged between −5.3% (±7.7%) at the upper limbs and −9.0% (±4.9%) at the trunk for FM (kg), and between −8.9% (±5.1%) at the lower limbs and −10.4% (±4.8 %) at the trunk for LTM (kg). At 12 months, all the parameters related to FM and LTM had significantly decreased, but the % relative variation in LTM was lower than in FM, which was also supported by an increase in the LTM/FM ratio. At 12 months, the reduction in % relative variation at the different sites ranged between −39.0% (±12.1%) at the lower limbs and −47.9 % (±14.1%) at the trunk for FM (kg), while the loss for LTM was about 20% and homogeneous between sites. No subsequent loss in FM or LTM was observed at 24 months. The prevalence of T2D, HTA and obstructive sleep apnoea decreased following surgery.

### 3.3. Sarcopenia Prevalence

ALM and ALM(h^2^) were significantly reduced, with a comparable percentage relative variation between presurgery and 1 month, and between 1 month and 12 months, around 10%. Conversely, ALMI(BMI) increased at 12 and 24 months in comparison with baseline values. Sarcopenia was diagnosed in 10 patients (4 men and 6 women) at baseline and in eight patients (4 men and 4 women) at 1 month only when the FNIH (ALMI(BMI)) criteria were used. Four patients (all women) and two patients (1 man and 1 woman) presented sarcopenia at 12 and 24 months, defined mainly by FNIH and EWGSOP (Table 2 and Figure 1).

### 3.4. Android, Gynoid and Abdominal Body Composition

All parameters (total mass, LTM and FM) measured at android or gynoid regions decreased significantly at 1 and 12 months and this was accentuated with the time from surgery, following the same pattern as observed for the whole body (Table 3 and Figure 1). This was characterized by an accentuated LTM at 1 month, while, for FM, the decrease was maintained over the first 12 months. At abdominal regions, a progressive FM loss (*p* < 0.001) for TAT (−7.69% at 1 month, −45.75% at 12 months), VAT (−7.16% at 1 month, −34.8% at 12 months) and SAT (−7.98% at 1 month, −46.89% at 12 months) was also observed. At the android, gynoid and abdominal regions, no subsequent loss was observed between 12 and 24 months.

### 3.5. Biological Parameters

The variation in biological parameters is presented in Table 4 and Figure 2. For glucose homeostasis, a significant decrease was observed in the levels of fasting glucose, HbA_1c_, insulin and HOMA-IR, suggesting improved insulin sensitivity at all postsurgical time points. No subsequent loss was observed for these biological parameters after 12 months. IGF-1 and IGF-BP3 decreased simultaneously after 1 month, whereas IGF-1 had higher values and IGFBP-3 had lower values at 12 and 24 months in comparison with presurgical values. The concentration of albumin did not change significantly with time, whereas a significant decrease in CRP at 1 month was observed and accentuated at 12 and 24 months.

Compared to baseline, absolute REEm significantly decreased by −15.7% at 1 month and continued to decrease at 12 months, which totalled a reduction of −23.2%. No subsequent decrease was observed between 12 and 24 months.

### 3.6. Correlations between Basal Parameters and Body Composition Change

The correlation analysis between the basal parameters and % relative body composition changes at the various time points is presented in Table 5. Few significant correlations were observed and only age seemed clearly associated with LTM and FM losses at 12 and 24 months.

## 4. Discussion

This study aimed to characterize the changes in body composition and biological and metabolic parameters in patients with obesity over a 24-month period following SG. The study demonstrated that a significant modification in body composition was associated with the body weight loss and was characterized by an acute reduction in LTM and a continuous loss of FM and VAT over the first 12 months. Concomitantly, an improvement in the lipid, glycaemic, inflammatory and somatotropic profiles was also demonstrated.

This longitudinal study, based on a cohort composed mostly of women, showed that SG induced a loss of approximately 29% of the presurgical body weight and 20% of the anthropometric parameters, including waist and hip circumferences, after 12 and 24 months. The positive effect of weight loss on glycaemic, lipid, inflammatory and somatotropic profiles was clearly confirmed even though some of the patients continued to be overweight or obese at 24-months postsurgery. Moreover, the LTM and FM losses were significant from the first month and maintained over the 2 years. However, the kinetics of loss seemed to be specific to the components. Thus, although the magnitude of variation was relatively comparable between FM and LTM in the first month—i.e., around 8–10%—almost half of the total loss in LTM observed in the first 12 months occurred during this acute phase. The loss in FM appeared more progressive and sustained during the subsequent prolonged weight-loss period, and at 12 months it largely exceeded that of LTM (−38 vs. −20%). It is interesting to note that the % relative variation in LTM and FM appeared relatively comparable between sites, including limbs, trunk and whole body throughout the study, suggesting a uniform body loss. These results are completely superimposable in terms of kinetics and intensity with those of two previously published studies by our group performed in a limited group of patients (*n* = 30) after the same follow-up periods [10,11]. The findings from our previous and current studies suggest that, for the same surgical procedure (i.e., SG) and patient characteristics (sex, age and BMI), the expected body composition changes are very reproducible. Comparison with other works must be made with caution because the type of surgical procedure (SG vs. RYGB vs. one-anastomosis gastric bypass) and the device used to measure body composition (BIA vs. DXA) may interfere with the results [6,13,14,33]. In a group of mainly female patients who had undergone predominantly SG, Sivakumar et al. [12] also reported that LTM depletion evaluated with DXA occurred predominantly in the initial month, whereas FM declined more progressively over the 12-month follow-up. A recent meta-analysis of 122 studies [13] highlighted that all types of BS cause LTM decline and, although the rate of LTM decrement decreases over time, it follows a significant downward trend over the first 12 months. It should be noted that no subgroup analysis according to the surgical procedure was performed in this meta-analysis [13]. Last, using BIA, two longitudinal studies [14,34] demonstrated that the greatest LTM loss (~7–10%) occurred between 1 and 3 months and that the loss slowed down at 12 months, with values of approximately 80−85% of the presurgical values.

The highest rate of LTM loss in the first months following BS may be explained by the reduction in physical activity, inadequate protein intake, and restricted global food intake during this period of about 700 kcal/day, which can promote proteolysis to meet the metabolic demands [35,36]. The initial decrease in IGF-1, an anabolic hormone that is sensitive to nutritional intake and even more so to protein intake, might confirm this hypothesis. In humans, weight decrease is associated with IGF-1 reduction only when energy intake is below 50% of the daily ration. [37,38]. In our study, we reported, for the first time, that, after 12 months, the IGF-1 values increased and exceeded the presurgical values. Functional hyposomatotropism is a situation frequently observed in obesity [39], and normalization of nutritional intake alone cannot explain this overcompensation. Obesity is also a disease characterized by the presence of low-grade chronic inflammation [40], and the recovery of normal IGF-1 synthesis over time after BS was reported to be independently related with CRP [39]. Consequently, the decrease in CRP found in our study might have led to an increase in IGF-1, which, in turn, might have reduced the rate of LTM loss after 1 month [41]. The favourable effect of IGF-1 in our study might even have been more effective because its free form was increased due to the reduction in its main binding protein (IGFBP-3) throughout the study. This finding has never before been reported. Ohira et al. [42] reported an important effect of IGF-1 on LTM because they found that preoperative values were related to maintaining skeletal muscle mass and decreasing body FM in a sex-dependent manner. Unfortunately, in this study [42], no information concerning the variation in IGF-1 after BS was reported. In line with our results, Juiz-Valina et al. [39] investigated 116 patients who had undergone RYGB or SG, reporting an initial decrease in IGF-1 after 1 month followed by a progressive increase in growth hormone and IGF-1 up to 12 months. Conversely, De Marinis et al. [43] studied 15 obese female patients 16–24 months after biliopancreatic diversion (BPD) and reported that, although the GH response to GH-releasing hormone markedly increased, the initially low IGF-1 and IGFBP-3 concentrations remained unchanged.

Very few studies have analysed body composition changes after weight stabilization over the 12 months following BS, and specifically SG [14,15,25,33,34]. Moreover, the data are generally drawn from cross-sectional analyses, which limits their scope [25,33]. Our results offer new findings by demonstrating that no substantial loss was observed for LTM and FM between 12 and 24 months, which suggests that a new steady state was established 12 months following BS. Two longitudinal studies of 24 and 36 months, using BIA, also observed minor body composition changes after 12 months following BS [14,34].

The reduction in body weight and FM after BS is a desired result, while the LTM loss may have deleterious functional and metabolic consequences. However, we observed, for the first time, that 24 months post-SG, the prevalence of low muscle mass—a criterion involved in the definition of sarcopenia evaluated by DXA—did not increase. This observation was made regardless of the threshold [ALM, ALMI(h^2^) or ALMI(BMI)]. The presurgical high LTM values in patients with obesity [44] and the maintenance of relative subnormal mean BMI (~28–29 kg/m^2^) after BS could be responsible. In line with our results, previous studies using DXA have reported that none of the patients presented pathologically low LTM from one to several years (2–18) after RYGB, thus ruling out the diagnosis of sarcopenia [25,45]. We note that the prevalence of sarcopenia may vary with the technique. For example, Vassilev et al. [46] used magnetic resonance imaging (RMI) and reported that 57% of the patients were sarcopenic after 24 weeks following RYGB, whereas Voican et al. [47] used computed tomography and observed a sarcopenia prevalence of 32% in patients 1 year after SG.

Although the LTM loss seems to have no or only a limited effect on the sarcopenia prevalence up to 24 months, it may alter the metabolic status because—as observed in previous studies as well as this one—muscle mass is the primary determinant of REE [19,34,48] rather than other parameters such as FM [34]. We observed that the main reductions in REE and LTM were concomitant at 1 month, with values continuing to decrease at 12 months and stabilized at 24 months. Two studies performed in patients with extreme obesity (BMI > 50 kg/m^2^) who underwent different surgical procedures [34,49] also reported this initial REE loss at 1 month that continued up to 24 months. However, it should be noted that body composition was not concomitantly evaluated [49] and that REE was indirectly evaluated by BIA [34], limiting the scope of these works. Our results clearly underlined that the first months, and particularly the first month, are crucial for LTM loss. Consequently, the implementation of programs in this period based on a nutritional approach, such as protein supplementation [50], or exercise interventions, particularly resistance training [51], should help preserve LTM and thus REE [52,53]. The maintenance of REE after BS is crucial and this problem needs to be resolved because low values are associated with weight regain [49,54]. The crucial role of LTM in weight homeostasis and metabolism point to the importance of evaluating body composition in each patient to improve weight loss [34]. However, as underlined by Martinez et al. [34], LTM is rarely evaluated in daily clinical practice.

In parallel to the conventional parameters of body composition obtained with DXA, we also evaluated the variation in abdominal adiposity. This approach may be more pertinent in this population because most of the obesity-related comorbidities are more related to VAT than whole body FM. For example, obesity, especially when linked to increases in visceral fat, has more deleterious effects on the cardiovascular system than subcutaneous compartments [22]. We demonstrated herein that abdominal FM parameters, including TAT, VAT and SAT, followed a profile similar to that of whole-body FM in terms of kinetics and intensity. Despite their pertinence, these parameters are largely under-analysed [10,24,55,56,57,58,59] and, to our knowledge, only two studies have evaluated them with comparable techniques (i.e., DXA) and surgical procedures [10,57], but with a limited follow-up duration. For example, Zang et al. [57] reported a single measurement point after only 3 months, with VAT decreased by approximately 22.5%. In our study, the relative variation in VAT was −7.8 at 1, −34.8 at 12, and −38.4 % at 24 months, and these results are totally in line with those previously published by our group at 1 and 12 months [10]. It is well established that diabetic patients have a greater amount of VAT before surgery compared to normoglycaemic subjects [24], and this observation persisted even later after surgery [25].

The identification of the basal parameters potentially associated with postsurgical body composition change is an attractive approach to implementing corrective measures. In this study, only age was positively but weakly correlated with LTM and FM loss at 12 and 24 months. Conversely, a recent study reported that the loss of LTM measured with BIA at 24 months was only associated with presurgical values of LTM and HOMA-IR independent of age, gender and BS techniques [34]. Other studies have reported that higher BMI and LTM before surgery, male gender and older age were associated with greater LTM loss [60]. Finally, higher preoperative BMI, female gender and patients undergoing SG were also found to be determinants of greater LTM [14,61]. Many factors, including gender and the age distribution of the studied group, may explain the discrepancies in the results.

## 5. Limitations and Strength

The strength of this longitudinal study is the long follow-up period after surgery (24 months) with four repeated measures from the acute loss of body weight up to body weight stabilization. Moreover, as recommended by the International Society for Clinical Densitometry (ISCD) guidelines [9], the body composition change after BS was evaluated by the gold standard technique (DXA). In addition, the concomitant evaluation of biological parameters reflecting glucose homeostasis, inflammation and metabolism may improve our knowledge on the potential link between body composition change and these parameters. We are aware that our study presents some limitations, including a limited number of patients at inclusion and after 24 months, due to the difficulties in maintaining patient compliance and the COVID-19 pandemic. However, the study group reflects the population managed in our surgical department, and the number of included patients is in line with previous studies [13], reflecting the difficulties of including and following these patients after BS. In addition, a longer study until weight regain occurs might provide more clues with factors that are closely associated with weight regain after SG. Moreover, our cohort was composed of men and women, and it cannot be excluded that a sex-specific body composition change before surgery occurred, particularly for VAT [59]. Our population was mainly women, and, thus, no subgroup analysis was performed. Last, specific data on food intake and physical activity levels were not collected. However, all patients had received the same hygienic and dietetic advice from a specialized team before and during the follow-up.

## 6. Conclusions

Our longitudinal study provides crucial findings on the kinetics and specificity of the two components of body composition change over a 2-year period after SG. FM tended to progressively decrease up to 12 months, whereas LTM loss was predominant in the first month. Taking into account the crucial role of LTM on REE regulation and potential weight regain, multidisciplinary (surgeons, dietitians, physiotherapists, …) reflection and dialogue should be encouraged to develop strategies for limiting LTM loss, particularly in the early postoperative period.

## Figures and Tables

**Figure 1 nutrients-15-01201-f001:**
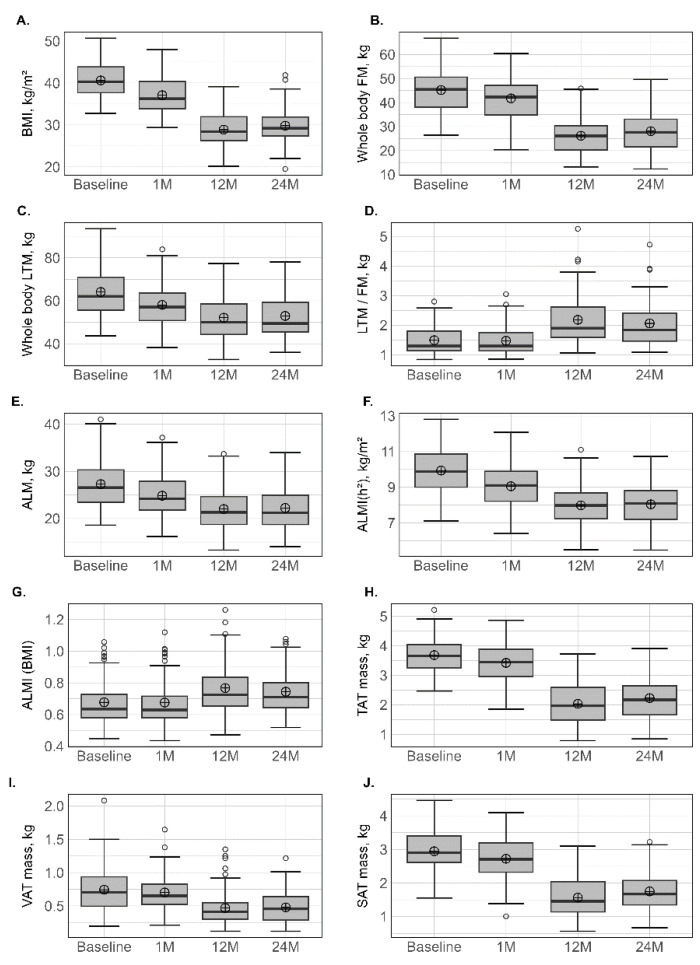
Variation in anthopometric and body composition parameters at different times. BMI: body mass index (**A**); FM: fat mass (**B**); LTM: lean tissue mass (**C**); LTM/FM ratio (**D**); ALM: appendicular lean mass (**E**); ALMI(h^2^): appendicular lean mass index (height^2^) (**F**); ALM/height^2^, ALMI(BMI) (**G**): appendicular lean mass index (BMI); TAT: total adipose tissue (**H**); VAT: visceral adipose tissue (**I**); SAT: subcutaneous adipose tissue (**J**). Lower whisker represents the smallest observation greater than or equal to lower hinge −1.5 × IQR. Upper whisker represents the largest observation less than or equal to upper hinge +1.5 × IQR. Lower and upper hinges represent, respectively, the 25 and the 75% quartile values. The open circles represent the mean value in each time. Data beyond the end of the whiskers are called outliers points and are plotted individually. IQR, interquartile range.

**Figure 2 nutrients-15-01201-f002:**
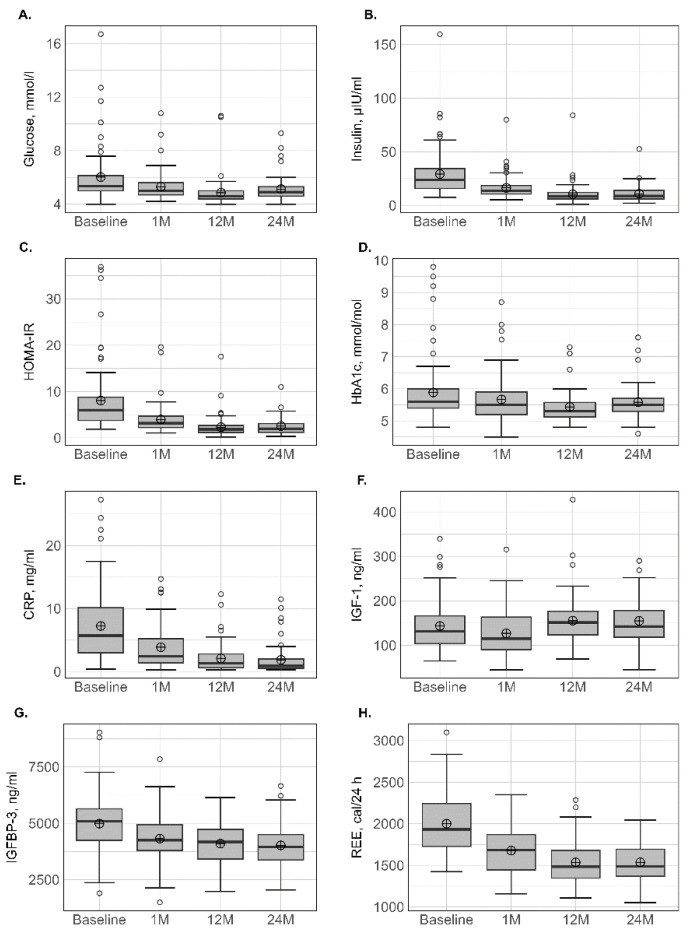
Variation in biological parameters at different times. BMI: body mass index (**A**); FM: fat mass (**B**); LTM: lean tissue mass (**C**); LTM/FM ratio (**D**); ALM: appendicular lean mass (**E**); ALMI(h^2^): appendicular lean mass index (height^2^) (**F**); ALM/height^2^, ALMI(BMI) (**G**): appendicular lean mass index (BMI); TAT: total adipose tissue (**H**); VAT: visceral adipose tissue (**I**); SAT: subcutaneous adipose tissue (**J**). Lower whisker represents the smallest observation greater than or equal to lower hinge −1.5 × IQR. Upper whisker represents the largest observation less than or equal to upper hinge + 1.5 × IQR. Lower and upper hinges represent, respectively, the 25 and the 75% quartile values. The open circles represent the mean value in each time. Data beyond the end of the whiskers are called outlier points and are plotted individually. IQR, interquartile range.

**Table 1 nutrients-15-01201-t001:** Clinical characteristics of the patients at baseline and 1, 12 and 24 months after sleeve gastrectomy.

	Baseline	1-Month	12-Months	24-Months	% Relative Variation (Δ 1 m-Baseline/Baseline)	% Relative Variation (Δ 12 m-Baseline/Baseline)	% Relative Variation (Δ 24 m-Baseline/Baseline)
Number of patients, n	83	83	76	60			
Age, yr	40.9 ± 12.3	-	-	-	-	-	-
Height, m	165.2 ± 7.6	-	-	-	-	-	-
Weight, kg	110.9 ± 13.0	100.9 ± 12.3	79.1 ± 14.2	81.9 ± 14.0	−9.1 ± 2.1 ***	−29.3 ± 8.4 ***	−27.5 ± 9.6 ***
BMI, kg/m^2^	40.7 ± 4.2	37.1 ± 4.2	28.9 ± 4.3	29.8 ± 4.5	−9.1 ± 2.1 ***	−29.3 ± 8.4 ***	−27.6 ± 9.8 ***
Ideal body weight, kg	60.00 ± 6.01	-	-	-	-	-	-
Neck circumference, cm	42.4 ± 4.0	40.1 ± 3.8	35.9 ± 3.6	36.4 ± 3.6	−4.8 ± 3.5 ***	−15.2 ± 4.7 ***	−15.0 ± 5.4 ***
Waist circumference, cm	114.2 ± 12.9	107 ± 11.4	89.3 ± 13.5	91.1 ± 11.8	−6.4 ± 5.1 ***	−21.6 ± 9.5 ***	−20.7 ± 8.9 ***
Hip circumference, cm	128.2 ± 11.7	121.8 ± 11.9	99.4 ± 15.6	103.9 ± 14.1	−5.0 ± 4.3 ***	−22.6 ± 11.0 ***	−19.3 ± 10.0 ***
Waist/hip circumference ratio	0.9 ± 0.1	0.9 ± 0.1	0.9 ± 0.1	0.9 ± 0.1	1.3 ± 7.7 **	2.5 ± 13.8	0.96 ± 11.1
Type 2 diabetes mellitus, n (%)	17 (20.48)	13 (15.85)	5 (6.67)	6 (9.68)			
Hypertension, n (%)	37 (44.58)	23 (28.75)	12 (15.58)	10 (13.70)			
Sleep apnoea, n (%)	53 (63.86)	34 (53.13)	25 (37.88)	9 (15.52)			

Data are presented as mean ± SD. BMI: body mass index. Δ 1 m-baseline/baseline represents the % relative difference between values at 1 month and baseline; Δ 12 m-baseline/baseline represents the % relative difference between values at 12 months and baseline; Δ 24 m-baseline/baseline represents the % relative difference between values at 24 months and baseline; the % relative variation was defined as [100 × (measure 2 − measure 1)/measure 1]; ** for *p* < 0.01 and *** for *p* < 0.001.

**Table 2 nutrients-15-01201-t002:** Whole body composition of the patients at baseline and 1, 12 and 24 months after sleeve gastrectomy.

	Baseline	1-Month	12-Months	24-Months	% Relative Variation (Δ 1 m-Baseline/Baseline)	% Relative Variation (Δ 12 m-Baseline/Baseline)	% Relative Variation (Δ 24 m-Baseline/Baseline)
Number of patients, n	83	83	76	60			
LTM (kg)							
Upper limb	3.3 ± 0.9	3.0 ± 8.4	2.7 ± 0.8	2.7 ± 0.8	−9.8 ± 5.9 ***	−20.6 ± 7.5 ***	−20.7 ± 7.5 ***
Trunk	33.6 ± 6.3	30.1 ± 5.6	27.0 ± 5.7	27.6 ± 5.5	−10.4 ± 4.8	−20.1 ± 7. 1 ***	−19.9 ± 7.9 ***
Lower limbs	10.3 ± 1.8	9.4 ± 1.7	8.3 ± 1.7	8.4 ± 1.6	−8.9 ± 5.1 ***	−20.2 ± 8.3 ***	−20.0 ± 8.0 ***
Whole body	64.3 ± 11.6	58.1 ± 10.4	52.9 ± 10.6	52.3 ± 10.4	−9.6 ± 4.0 ***	−19.4 ± 6.5 ***	−19.5 ± 7.1 **
FM (kg)							
Upper limbs	2.9 ± 0.7	2.7 ± 0.7	1.7 ± 0.6	1.8 ± 0.6	−5.3 ± 7.7 ***	−39.8 ± 13.9 ***	−38.7 ± 15.3 ***
Trunk	21.8 ± 4.4	19.8 ± 4.1	11.6 ± 4.4	12.8 ± 7.3	−9.0 ± 4.9 ***	−47.9 ± 14.1 ***	−42.3 ± 16.4 ***
Lower limbs	8.2 ± 2.5	7.6 ± 2.3	5.0 ± 1.7	5.3 ± 1.8	−7.2 ± 3.7 ***	−39.0 ± 12.1 ***	−34.8 ± 14.1 ***
Whole body	45.2 ± 9.2	41.7 ± 8.9	26.2 ± 8.1	28.1 ± 8.4	−7.9 ± 3.5 ***	−42.7 ± 12.5 ***	−38.2 ± 14.6 ***
FM (%)							
Upper limbs	45.4 ± 9.8	46.4 ± 10.0	38.2 ± 10.1	38.3 ± 10.2	−2.3 ± 4.6 ***	−16.6 ± 9.8 ***	−15.4 ± 11.6 ***
Trunk	39.0 ± 5.8	39.3 ± 6.1	29.0 ± 7.1	30.5 ± 7.3	0.6 ± 5.4	−26.0 ± 12.7 ***	−21.0 ± 14.9 ***
Lower limbs	42.4 ± 9.5	43.1 ± 9.5	35.7 ± 9.1	37.1 ± 9.5	1.5 ± 5.8 *	−16.3 ± 9.5 ***	-12.2 ± 9.3 **
Whole body	40.4 ± 7.0	40.8 ± 7.2	32.2 ± 7.2	33.4 ± 7.5	0.8 ± 4.1	−20.8 ± 9.9 ***	−16.7 ± 11.6 ***
LTM/FM							
Upper limbs	1.3 ± 0.6	1.2 ± 0.6	1.7 ± 0.9	1.7 ± 0.9	−4.3 ± 9.0 ***	37.7 ± 29.7 ***	37.7 ± 38.7 ***
Trunk	1.6 ± 0.4	1.6 ± 0.4	2.6 ± 1.0	2.4 ±1.0	−1.1 ± 8.7	64.1± 46.7 ***	52.1 ± 53.7 ***
Lower limbs	1.4 ± 0.7	1.4 ± 0.7	1.9 ± 1.0	1.8 ± 0.9	−1.7 ± 7.4 *	35.2 ± 27.5 ***	27.6 ± 26.1 ***
Whole body	1.5 ± 0.5	1.5 ± 0.5	2.2 ± 0.8	2.1 ± 0.8	−1.6 ± 6.9 *	46.4 ± 30.6 ***	37.7 ± 35.5 ***
Sarcopenia index							
ALM, kg	27.40 ± 5.33	24.84 ± 4.85	22.02 ± 4.97	22.25 ± 4.74	−9.2 ± 4.6 ***	−20.3 ± 7.7 ***	−20.2 ± 7.6 ***
ALMI(h^2^), kg/m^2^	9.97 ± 1.29	9.05 ± 1.17	7.98 ± 1.21	8.04 ± 1.16	−9.2 ± 4.6 ***	−20.3 ± 7.7 ***	−20.3 ± 7.6 ***
ALMI(BMI)	0.68 ± 0.14	0.68 ± 0.15	0.77 ± 0.16	0.74 ± 0.15	−0.2 ± 4.0	13.3 ± 8.9 ***	10.8 ± 9.0 ***

Data are presented as mean ± SD. LTM: lean tissue mass; FM: fat mass; ALM: appendicular lean mass was defined as the sum of the LTM of the arms and legs; ALMI(h^2^): appendicular lean mass index (height^2^) was defined as ALM/height^2^, ALMI(BMI): appendicular lean mass index (BMI) was defined as ALM/BMI; Δ 1 m-baseline/baseline represents the % relative difference between values at 1 month and baseline; Δ 12 m-baseline/baseline represents the % relative difference between values at 12 months and baseline; Δ 24 m-baseline/baseline represents the % relative difference between values at 24 months and baseline; the % relative variation was defined as [100 × (measure 2 − measure 1)/measure 1]; * indicates a significant variation for *p* < 0.05, ** for *p* < 0.01 and *** for *p* < 0.001.

**Table 3 nutrients-15-01201-t003:** Android, gynoid and abdominal adipose tissue at baseline and 1, 12 and 24 months after sleeve gastrectomy.

	Baseline	1-Month	12-Months	24-Months	% Relative Variation (Δ 1 m-Baseline/Baseline)	% Relative Variation (Δ 12 m-Baseline/Baseline)	% Relative Variation (Δ 24 m-Baseline/Baseline)
Number of patients, n	83	83	76	60			
Android region							
Total mass, kg	9.9 ± 1.7	8.7 ± 1.5	6.3 ± 1.8	6.6 ± 1.8	−11.8 ± 4.3 ***	−36.6 ± 10.8 ***	−34.7 ± 12.0 ***
LTM, kg	5.7 ± 1.1	5.0 ± 1.0	4.3 ± 1.1	4.4 ± 1.0	−12.9 ± 5.8 ***	−24.5± 8.9 ***	−24.8 ± 10.2 ***
FM, kg	4.2 ± 0.9	3.7 ± 0.9	2.0 ± 0.9	2.2 ± 0.9	−10.4 ± 7.2 ***	−53.2 ± 15.3 ***	−48.2 ± 17.6 ***
Fat, %	42.1 ± 5.4	42.8 ± 6.0	30.5 ± 7.1	32.3 ± 7.6	1.7 ± 6.6 *	−27.7 ± 13.7 ***	−22.7 ± 16.4 ***
Gynoid region							
Total mass, kg	18.6 ± 2.6	16.9 ± 2.4	13.4 ± 2.5	13.9 ± 2.5	−9.0 ± 4.9 ***	−28.2 ± 10.3 ***	−25.9 ± 11.5 ***
LTM, kg	11.2 ± 1.9	10.0 ± 1.7	8.9 ± 1.7	9.0 ± 1.7	−10.6 ± 6.1 ***	−21.0 ± 9.0 ***	−21.3 ± 10.7 ***
FM, kg	7.4 ± 2.0	6.9 ± 1.8	4.5 ± 1.4	4.9 ± 1.6	−6.3 ± 6.1 ***	−38.5 ± 13.2 ***	−32.2 ± 15.1 ***
Fat, %	39.3 ± 8.2	40.7 ± 8.0	33.3 ± 7.3	35.1 ± 7.6	3.0 ± 5.2 ***	−15.1 ± 8.3 ***	−9.2 ± 10.9 ***
Abdominal adipose tissue							
TAT mass, kg	3.67 ± 0.59	3.42 ± 0.63	2.03 ± 0.71	2.23 ± 0.75	−7.69 ± 5.73 ***	−45.75 ± 14.74 ***	−40.33 ± 16.67 ***
VAT mass, kg	0.74 ± 0.33	0.70 ± 0.26	0.47 ± 0.26	0.48 ± 0.24	−7.16 ± 18.45 ***	−34.80 ± 35.12 ***	−38.40 ± 26.08 ***
SAT mass, kg	2.95 ± 0.61	2.72 ± 0.60	1.57 ± 0.62	1.75 ± 0.63	−7.98 ± 8.36 ***	−46.89 ± 18.23 ***	−39.40 ± 19.93 ***

Data are presented as mean ± SD. Android: waist and abdomen area, gynoid: hip area, TAT: total adipose tissue; VAT: visceral adipose tissue; SAT: subcutaneous adipose tissue. Δ 1-m baseline/baseline represents the % relative difference between values at 1 month and baseline; Δ 12-m baseline/baseline represents the % relative difference between values at 12 months and baseline; Δ 24-m baseline/baseline represents the % relative difference between values at 24 months and baseline; the % relative variation was defined as [100 × (measure 2 − measure 1)/measure 1]; * indicates a significant variation for *p* < 0.05, and *** for *p* < 0.001.

**Table 4 nutrients-15-01201-t004:** Biological parameters at baseline and 1, 12 and 24 months after sleeve gastrectomy.

	Baseline	1-Month	12-Months	24-Months	% Relative Variation (Δ 1-m Baseline/Baseline)	% Relative Variation (Δ 12-m Baseline/Baseline)	% Relative Variation (Δ 24-m Baseline/Baseline)
Number of patients, n	83	83	76	60			
Glucose homeostasis							
Glucose, mmol/l	6.1 ± 2.1	5.3 ± 1.0	4.9 ± 1.1	5.1 ± 0.9	−8.2 ± 13.9 ***	−15.6 ± 14.2 ***	−15.3 ± 13.7 ***
Insulin, µIU/mL	29.3 ± 22.0	16.3 ± 10.6	10.4 ± 10.2	10.8 ± 7.8	−34.2 ± 37.4 ***	−61.4 ± 22.8 ***	−59.2 ± 24.7 ***
HOMA−IR	8.1 ± 7.1	4.0 ± 3.1	2.3 ± 2.3	2.5 ± 1.9	−38.1 ± 39.0	−66.2 ± 23.8	−65.5 ± 21.6
HbA1c, %	5.9 ± 1.0	5.7 ± 0.7	5.4 ± 0.5	5.6 ± 0.6	−3.3 ± 4.5 ***	−6.4 ± 8.4 ***	−5.9 ± 9.3 ***
HbA1c, mmol/mol	41.3 ± 10.7	38.3 ± 7.7	35.8 ± 5.4	37.5 ± 6.7	−5.8 ± 6.7 ***	−11.0 ± 11.0 ***	−9.4 ± 12.0
Lipid profile							
Total cholesterol, g/l	2.0 ± 0.5	1.8 ± 0.4	1.9 ± 0.6	1.9 ± 0.3	−7.3 ± 20.0 ***	3.3 ± 31.6	0.3 ± 16.6
HDL, g/l	0.5 ± 0.1	0.5 ± 0.1	0.6 ± 0.2	0.6 ± 0.1	−5.5 ± 17.1 ***	28.9 ± 33.9	32.1 ± 24.7 ***
LDL, g/l	1.2 ± 0.4	1.0 ± 0.3	1.1 ± 0.3	1.1 ± 0.3	−10.1 ± 20.4 ***	−1.6 ±29.3 ***	−0.5 ± 35.4
Triglycerides, g/l	1.6 ± 0.8	1.4 ± 0.6	1.0 ± 0.4	0.9 ± 0.4	−2.3 ± 31.0 (*p* = 0.06)	−28.9 ± 26.2 ***	−30.3 ± 25.7 ***
Other parameters							
CRP, mg/ml	7.3 ± 5.7	3.8 ± 3.4	2.1 ± 2.2	1.9 ± 2.4	−36.4 ± 42.7 ***	−66.4 ± 32.4 ***	−64.7 ± 55.6 ***
IGF-1, ng/ml	143.6 ± 55.5	127.0 ± 51. 9	155.5 ± 55.5	155.0 ± 52.5	−10.8 ± 24.1 ***	12.5 ± 23.3 ***	16.0 ± 29.7 ***
IGFBP-3, ng/ml	4972.5 ± 1324.0	4321.8 ± 1140.14	4099.6 ± 958.0	4022.6 ± 961.0	−12.3 ± 14.4 ***	−15.3 ± 11.8 ***	−12.9 ± 16.7 ***
Albumin, g/l	44.5 ± 2.9	44.5 ± 4.3	45.6 ± 6.4	45.1 ± 4.3	−1.1 ± 7.2	−0.8 ± 7.3	1.2 ± 8.5
REE, cal/24 h	2010.2 ± 366.6	1677.8 ± 282.6	1535.6 ± 253.4	1536.5 ± 239.6	−15.7 ± 10.0 ***	−23.2 ± 8.3 ***	−24.0 ± 9.3 ***
Predicted REE values, %	4.5 ± 12.9	−6.5 ± 11.7	−3.0 ± 8.5	−3.5 ± 9.0	-	-	-

Data are presented as mean ± SD. HOMA-IR: homeostasis model assessment of insulin resistance; HbA1c: glycated haemoglobin; HDL: high- density lipoprotein; LDL: low-density lipoprotein; CRP: C-reactive protein; IFG-1: insulin-like growth factor-1; IGFBP-3: insulin-like growth factor binding protein-3; REE: resting energy expenditure; Predicted REE: resting energy expenditure predicted from the equation of Harris and Benedict modified by Roza and Shizgal [32]. Δ 1-m baseline/baseline represents the % relative difference between values at 1 month and baseline; Δ 12-m baseline/baseline represents the % relative difference between values at 12 months and baseline; Δ 24-m baseline/baseline represents the % relative difference between values at 24 months and baseline; the % relative variation was defined as [100 × (measure 2 − measure 1)/measure 1]; *** for *p* < 0.001.

**Table 5 nutrients-15-01201-t005:** Correlation between preoperative characteristics and basal or variations in body composition parameters.

	Baseline					% Relative Variation (Δ 1 m-Baseline/Baseline)					% Relative Variation (Δ 12 m-Baseline/Baseline)					% Relative Variation (Δ 24 m-Baseline/Baseline)				
Baseline Parameters	LTM	FM	SAT	VAT	TAT	LTM	FM	SAT	VAT	TAT	LTM	FM	SAT	VAT	TAT	LTM	FM	SAT	VAT	TAT
Age	0.12	−0.30 **	−0.44 ***	0.64 ***	−0.09	0.10	−0.09	−0.09	−0.30 **	−0.19	0.29 *	0.35 **	0.22	0.12	0.25	0.26 *	0.29 *	0.35 **	0.02	0.31 *
BMI	0.22 *	0.79 ***	0.63 ***	0.17	0.72 ***	0.26 *	−0.03	0.14	0.11	0.19	−0.05	0.04	0.01	0.06	0.13	−0.2	−0.10	0.01	−0.23	−0.07
WB LTM	−	−0.19	−0.26	0.36	−0.08	−0.10	−0.27 *	−0.08	−0.33 **	−0.27 *	0.05	0.15	0.16	0.07	0.15	−0.02	0.07	0.06	−0.08	0.04
WB FM	−0.19	−	0.81***	−0.02	0.83 ***	0.34 **	0.11	0.24	0.22	0.34 **	0.03	0.07	0.14	0.10	0.18	−0.1	−0.09	0.05	−0.18	−0.02
TAT	−0.08	0.83 ***	0.83 ***	0.25 *	-	0.38 ***	0.07	0.23 *	0.03	0.25 *	0.13	0.22	0.29 *	0.12	0.33 **	−0.02	0.08	0.22	−0.14	0.15
VAT	0.36 **	−0.02	−0.27 *	-	0.24 *	0.17	−0.04	0.18	−0.55 ***	−0.06	0.30**	0.45***	0.43***	−0.03	0.42***	0.21	0.39 *	0.48 ***	−0.10	0.41 **
SAT	−0.26 *	0.81 ***	-	−0.27*	0.83 ***	0.30 **	0.08	0.05	0.30 **	0.23 *	0.015	−0.02	0.02	0.14	0.08	−0.07	−0.13	−0.09	−0.06	−0.09
Glucose	0.45 ***	−0.22*	−0.32 **	0.50 ***	−0.06	0.08	−0.16	−0.09	−0.36 **	−0.24 *	0.12	0.20	0.18	−0.05	0.16	0.04	0.21	0.28 *	0.04	0.25
Insulin	0.37 ***	−0.03	−0.09*	0.20 *	0.01	−0.06	0.03	0.10	−0.09	0.07	−0.24 *	0.08	0.16	−0.1	0.13	−0.04	0031	0.28 *	0.18	0.30
HOMA-IR	0.47 ***	−0.12	−0.22 *	0.35 **	−0.01	0.03	−0.07	0.07	−0.23 *	−0.03	−0.18	0.13	0.21	−0.2	0.16	0.01	0.36	0.38 **	0.14	0.37
HbA1c	0.40 ***	−0.18	−0.32 **	0.53 ***	−0.02	0.13	−0.07	0.07	−0.33 **	−0.06	0.09	0.25 *	0.25 *	0.05	0.24 *	0.01	0.28	0.37 **	0.06	0.33 *
Total cholesterol	−0.12	−0.26 *	−0.16	0.11 *	−0.08	−0.16	0.02	−0.09	0.04	0.03	0.02	−0.06	−0.05	−0.07	−0.07	0.09	0.02	0.05	0.09	0.1
HDL	−0.41 ***	0.01	0.12	−0.30 **	−0.04	−0.09	0.23 *	0.02	0.20	0.16	0.05	−0.04	0.01	0.18	−0.01	0.03	−0.16	−0.15	0.13	−0.10
LDL	−0.15	−0.14	−0.09	0.05	−0.06	−0.09	−0.04	−0.08	0.07	−0.02	0.02	−0.1	−0.12	−0.08	−0.11	0.08	−0.02	0.02	0.10	0.04
Triglycerides	0.24 *	−0.26 *	−0.27 *	0.39 ***	−0.01	−0.05	−0.06	0.10	−0.28 *	−0.14	0.01	0.16	0.13	−0.04	0.11	0.01	0.26	0.29	0.01	0.28 *
CRP	−0.30 **	0.36 **	0.32 **	−0.12	0.24 *	0.20	0.24	0.10	0.20	0.18	0.05	0.12	0.07	0.25 *	0.18	0.01	−0.09	0.02	−0.04	−0.03
IGF-1	−0.01	0.04	0.19	−0.49 ***	−0.06	−0.06	−0.01	−0.15	0.17	−0.09	−0.08	−0.27 *	−0.25 *	−0.06	−0.28 *	−0.04	−0.20	−0.24	0.01	−0.21
IGFBP-3	0.08	0.10	0.19	−0.16	0.13	−0.07	0.04	0.06	0.14	0.12	−0.26	−0.21	−0.12	−0.04	−0.16	−0.18	−0.23	−0.11	−0.38*	−0.16
Albumin	0.42 ***	−0.34 **	−0.23*	−0.01	−0.23*	−0.10	−0.31 **	−0.20	−0.21 *	−0.28*	−0.14	−0.10	−0.04	−0.09	−0.11	−0.11	0.05	−0.03	0.02	0.01
REE	0.84 ***	−0.09	−0.17	0.37***	0.03	0.06	−0.25 *	0.01	−0.36 ***	−0.18	0.08	−0.18	0.22	0.01	0.19	0.01	0.15	0.17	−0.06	0.14

Data are presented as Spearman or Pearson correlation coefficients. * indicates a significant variation for *p* < 0.05, ** for *p* < 0.01, *** for *p* < 0.001. HOMA-IR: homeostasis model assessment of insulin resistance; HbA1c: glycated haemoglobin; HDL: high-density lipoprotein; LDL: low-density lipoprotein; CRP: C-reactive protein; IFG-1: insulin-like growth factor-1; IGFBP-3: insulin-like growth factor binding protein-3; REE: resting energy expenditure.

## Data Availability

The data used in the present analysis can be obtained through request to the corresponding author.

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
