# Peer review of "Changes in Lean Tissue Mass, Fat Mass, Biological Parameters and Resting Energy Expenditure over 24 Months Following Sleeve Gastrectomy"

_nutrients, 2023, doi:10.3390/nu15051201_

Round 1
Reviewer 1 Report
This is a well written study that analyzes lean muscle mass and fat mass in patients that have had a sleeve gastrectomy over 24 months. The data is presented well and the conclusions support the findings. The study is novel in that it follows patients for a variety of parameters over 24 months.
Author Response
Dear reviewer, thank you very much for your very complimentary comments. They are like a crown to a massive work that brought together many teams.

Reviewer 2 Report
In this study, the authors followed up the patients with sleeve gastrectomy, and did the longitudinal studies of biological parameters related to metabolism and inflammation, which did provide detailed information in the first 24 months. However, there are already similar studies with different following up time periods, most of the data this study is expected, and doesn't provide more useful insights. I would suggest the authors do another 12 months follow up when weight regain occurs, which might provide more clues with factors that are closely associated with weight regain after SG. Another weakness is the limited participants.
Major points:
1. There are already published studies with similar design but different follow up time, the authors should extract more useful information from this study that other studies don't have.
2. The authors should note the participants left the study whether are type 2 diabetes or not, since there are only 17 participants in this study, this information may change the ratio in Table 1.
3. Graph provides more straightforward information, the authors should present their important data with graph.
Minor:
There are several typos in the manuscript. There should be space between numbers and units.
Author Response
Dear Reviewer, thank you for the pertinent comments. We have taken into account all comments and all the modifications have been underlined in the revised manuscript.
Reviewer 2
In this study, the authors followed up the patients with sleeve gastrectomy, and did the longitudinal studies of biological parameters related to metabolism and inflammation, which did provide detailed information in the first 24 months. However, there are already similar studies with different following up time periods, most of the data this study is expected, and doesn't provide more useful insights. I would suggest the authors do another 12 months follow up when weight regain occurs, which might provide more clues with factors that are closely associated with weight regain after SG. Another weakness is the limited participants.
Thank you for these pertinent comments. We are quite aware that several studies have evaluated body composition changes after bariatric surgery. However, when an exhaustive analysis of these studies was performed, it appeared that most had focused on patients with RYGB, body composition was analysed with bioelectrical impedance analysis (BIA), and most reported a mean follow-up duration of 12 months or less (please see for the reference the systematic review and meta-analysis of 122 studies and 10,758 participants published recently by {Haghighat, 2022 #1941}. To the best of our knowledge, it appears that only five studies reported on a follow-up of longer than 12 months:
|
Study |
Type of study |
Time of investigation |
Number of patients |
Surgery Type |
Methods of body composition evaluation |
|
Barzin et al., {Barzin, 2021 #1953} |
Longitudinal study |
Baseline 1,3,6,9,12,18,24,36 months |
3864 |
2746 SG 1118 OAGB |
BIA |
|
Buhler et al., {Buhler, 2021 #1969} |
Cross-sectional |
Baseline and after mean 6.7 years |
142 |
SG + RYGB |
DXA |
|
Martinez et al. {Martinez, 2022 #1950} |
Longitudinal |
Baseline, 1, 6 12, 24 months |
85 |
RYGB + SG |
BIA |
|
Santini et al., {Santini, 2022 #1945} |
Cross-sectional |
Baseline and after duration > 24 months |
60 postmenopausal women |
RYGB |
DXA |
|
Sherf-Dagan et al., {Sherf-Dagan, 2019 #1645} |
Longitudinal |
Baseline, 1, 3, 6, 12, 18,30, 36 months |
60 |
SG |
BIA |
SG: Sleeve gastrectomy; RYGB: Roux-en-Y gastric bypass; OAGB: one-anastomosis gastric bypass ()
According to the comments, we have added several sentences to the introduction that clarify the originality of our study:
“Generally, the time of follow-up in these studies has been limited to the first 12 months [11-13], and very few longitudinal studies have been performed between 24 and 36 months [14,15], even though monitoring body composition change over a longer postsurgery period may provide clinically relevant information to better identify and treat patients according to lifestyle and medical care [13,16]. Moreover, although it is not clear whether body composition changes vary with the surgical procedure, studies have preferentially focused on Roux-en-Y gastric bypass (RYGB) rather than sleeve gastrectomy (SG) [13], despite SG recently becoming the most common bariatric approaches [17]”.
“However, to the best of our knowledge, few data have investigated the VAT change after BS [24,25]”.
Concerning the duration of follow-up, we totally agree with the comments about including another 12 or 24 months of follow-up in the period in which weight regain occurs, as this may give new crucial information. However, due to human and financial limitations and the COVID pandemic, we had to stop the exhaustive follow-up (DXA, IGF-1 and IGFBP-3 assays, and resting energy expenditure) for these patients at 24 months. We are very sorry. Nevertheless, to highlight this limitation, we have introduced in the limitations a sentence inspired by your suggestions: “In addition, a longer study until weight regain occurs might provide more clues with factors that are closely associated with weight regain after SG”.
Concerning the number of patients included in this study, we are aware that 83 patients at the inclusion may be considered as limited. However, this group reflects the patients seen in our surgical department and the number of patients included is in agreement with previously published studies {Haghighat, 2022 #1941}. After 24 months, we were still able to follow 60 patients despite the observation that compliance for medical follow-up in bariatric surgery patients is poor. Moreover, the COVID-19 pandemic increased the number of patients who were lost to follow-up. To clearly state that the number of patients was limited, we introduce the following sentences in the Limitations and Strength part: “We are aware that our study presents some limitations, including a limited number of patients at inclusion and after 24 months, due to the difficulties in maintaining patient compliance and the COVID-19 pandemic. However, the study group reflects the population managed in our surgical department, and the number of included patients is in line with previous studies {Haghighat, 2022 #1941}, reflecting the difficulties of including and following these patients after bariatric surgery.”
Major points:
- There are already published studies with similar design but different follow up time, the authors should extract more useful information from this study that other studies don't have.
Thank you for this pertinent comment. We have carefully compared our data with the literature, and wherever possible we have highlighted the new data from our study. Thank you again for your comment, which has helped us to significantly improve the quality of our study.
The following sentences have been introduced throughout the manuscript:
The comparison with other works must be made with caution because the type of surgical procedure (SG vs RYGB vs one-anastomosis gastric bypass) and the device used to measure body composition (BIA vs DXA) may interfere with the results [6,13,14,33].
In our study, we reported for the first time that after 12 months, the IGF-1 values increased and exceeded the presurgical values.
The favourable effect of IGF-1 in our study might even have been more effective because its free form was increased due to the reduction in its main binding protein (IGFBP-3) throughout the study. This finding has never before been reported.
Very few studies have analysed the body composition change after weight stabilization over the 12 months following BS, specifically SG [14,15,25,33,34].
Moreover, the data are generally drawn from cross-sectional analyses, which limits their scope [25,33]. Our results offer new findings by demonstrating that no substantial loss was observed for LTM and FM between 12 and 24 months, which suggests that a new steady state was established 12 months following BS.
The reduction in body weight and FM after BS is a desired result, while the LTM loss may have deleterious functional and metabolic consequences. However, we observed for the first time that 24 months post-SG, the prevalence of low muscle mass—a criterion involved in the definition of sarcopenia evaluated by DXA—did not increase.
However, it should be noted that body composition was not concomitantly evaluated [49] or that RMR was indirectly evaluated by BIA [34]
Despite their pertinence, these parameters are largely under-analysed [10,24,55−59], and to our knowledge only two studies have evaluated them with comparable techniques (i.e., DXA) and surgical procedures [10,57], but with a limited follow-up duration.
- The authors should note the participants left the study whether are type 2 diabetes or not, since there are only 17 participants in this study, this information may change the ratio in Table 1.
Thank you for this comment. We reanalysed the data and among the patients that presented T2D at baseline (n=17), only two of them were lost of follow-up. This information has been added in the results. “All the comorbidities decreased with time. Specifically, T2D was only present in six patients (<10%) at 24 months. Among the 17 patients that presented T2D at baseline, only two of them were lost to follow-up (one was died).
- Graph provides more straightforward information, the authors should present their important data with graph.
Thank you for the comments. As requested, the most important data have been presented with figures 1 and 2.
Minor:
There are several typos in the manuscript. There should be space between numbers and units.
Thanks for the comment, the article has been proofread by a native English and the manuscript had been corrected according to the editorial recommendations of Nutrients.

Round 2
Reviewer 2 Report
The authors made great efforts to unique their research, and the revision is pretty satisfying. They will need to provide high resolution graphs.